# Fabrication of Highly Transparent Y$_2$O$_3$ Ceramics via Colloidal Processing Using ZrO$_2$-Coated Y$_2$O$_3$ Nanoparticles

**Zhongchao Fu** [1,*]**, Nan Wu** [1]**, Haibo Long** [1]**, Jianming Wang** [2]**, Jun Zhang** [3]**, Zhaoxia Hou** [1]**, Xiaodong Li** [4]
**and Xudong Sun** [4]

1    Liaoning Province Key Laboratory of Micro-Nano Materials Research and Development,
     School of Mechanical Engineering, Shenyang University, Shenyang 110044, China;
     wunan20102010@163.com (N.W.); flydragon0109@163.com (H.L.); luckyxia2007@126.com (Z.H.)
2    Liaoning Key Laboratory of Advanced Materials Preparation Technology, School of Mechanical Engineering,
     Shenyang University, Shenyang 110044, China; wjmlucky1979@163.com
3    Key Laboratory of Research and Application of Multiple Hard Films, College of Mechanical Engineering,
     Shenyang University, Shenyang 110044, China; zhjun88@126.com
4    Key Laboratory for Anisotropy and Texture of Materials (Ministry of Education), Northeastern University,
     Shenyang 110819, China; xdli@mail.neu.edu.cn (X.L.); xdsun@mail.neu.edu.cn (X.S.)
*    Correspondence: zhongchaofu@syu.edu.cn; Tel.: +86-024-6226-9467

**Abstract:** An easy approach is described for the preparation of ZrO$_2$-coated Y$_2$O$_3$ nanopowder from a solution of zirconium nitrate with commercial Y$_2$O$_3$ nanopowder. The evolution process of the ZrO$_2$ coating layer upon calcination, such as the phase and microstructure of the particles' surface, was studied. Calcination of the powder at 700 °C resulted in ZrO$_2$-coated Y$_2$O$_3$ nanopowder. The rheological properties of the suspensions of ZrO$_2$-coated Y$_2$O$_3$ powders were studied. A well-dispersed suspension with a solid loading of 35.0 vol% using ZrO$_2$-coated Y$_2$O$_3$ nanopowder was obtained. The consolidated green body obtained by the centrifugal casting method showed improved homogeneity with a relative density of 50.2%. Transparent ceramic with high transparency and an average grain size of 1.7 μm was obtained by presintering at 1500 °C for 16 h in air, followed by post-HIP at 1550 °C for 2 h under 200 MPa pressure. The in-line transmittance at the wavelength of 1100 nm (1.0 mm thick) reached 81.4%, close to the theoretical transmittance of Y$_2$O$_3$ crystal.

**Keywords:** Y$_2$O$_3$; ZrO$_2$; transparent ceramics; colloidal processing; nanopowders





## 1. Introduction

Transparent yttria (Y$_2$O$_3$) is one of the most promising materials for application in high-power lasers [1], heat-resistive transparent windows [2], high-temperature refractories, and semiconductor devices [3], owing to its broad wavelength range of transparency (0.2–8.0 μm), high corrosion resistance, high melting point (2430 °C), high thermal conductivity, low phonon energy, and easy doping of rare-earth activator ions with a band gap energy of 6.3 eV [4]. To achieve fully densified ceramics with desired optical properties, fabrication of polycrystalline Y$_2$O$_3$ ceramics requires careful control of each step of ceramic processing, including powder synthesis, consolidation, and sintering. Simultaneously, each step has a direct impact on the following step.

By utilization of Y$_2$O$_3$ powder produced via established synthesis routes or directly using commercial powder, sintering methods and mechanisms have been mostly studied by researchers recently [5–9]. Typically, transparent Y$_2$O$_3$ ceramics can be sintered by pressure-assisted sintering processes such as spark plasma sintering (SPS) [5], hot pressing (HP) [6], and hot isostatic pressing (HIP) [7]. Pressureless sintering methods, such as vacuum sintering [10] and hydrogen atmosphere sintering [11], are also effective. Moreover, the HIP sintering method utilizes three-dimensional gas pressure to presintered bodies that are in the final stage of sintering, which is much more effective to eliminate pores compared

to other sintering methods. Presintering in air followed by HIP treatment possesses more advantages than traditional vacuum presintering plus HIP treatment, including energy-saving, low cost, and efficient process without annealing [12]. On the other hand, fully dense $Y_2O_3$ ceramics can be obtained using $ZrO_2$ as sintering additives, which can decrease the grain boundary mobility, leading to $Y_2O_3$ ceramics with a fully dense fine microstructure and high optical properties [13]. According to the published literature, fabrication of highly transparent ceramics has mainly been achieved by vacuum presintering plus HIP treatment, which requires high-cost equipment and a long annealing process. Compared with vacuum presintering, presintering in air is a low-cost and efficient process without annealing. In 2015, Wang et al. [7] fabricated transparent $Y_2O_3$ ceramics through air presintering at 1550 °C plus HIPing at 1600 °C for 3 h. The in-line transmittance of the ceramic was 81.7% at 1064 nm, and the thickness of the sample was 1.0 mm. In 2016, Liu et al. [8] also reported transparent $Y_2O_3$ ceramics by air presintering plus HIPing. However, the sample only showed an IR transmittance of 83.0% in the 3.0 to 5.0 μm wavelength region

It is well known that colloidal processing has exhibited advantages over traditional dry pressing by producing green compacts with a homogenous microstructure, high relative density, and more versatile geometrical options [14–16]. There are, however, to the best of our knowledge, relatively few studies devoted to the colloidal processing of $Y_2O_3$ green bodies. $Y_2O_3$ powders are highly reactive in aqueous medium. Dissolving ions, including $Y^{3+}$, $Y(OH)^{2+}$, and $Y_2(OH)_2^{4+}$, which can suppress the electrical double layer, is the most critical challenge in preparing long-term well-dispersed suspensions with high solids loading [17–20]. In order to inhibit the hydrolysis issue of $Y_2O_3$ during colloidal processing, in 2014, Sun et al. [20] coated $Y_2O_3$ powder by a nucleophilic addition reaction between polyurethane and tetraethylene pentamine. In 2017, Xu et al. [21] prepared $Y_2O_3$ suspensions using alcohol as the dispersion medium. However, the boiling point of ethanol was as low as 78 °C, and the consolidated green compacts were easy to crack during the drying process. Until now, the solid loading of $Y_2O_3$ suspensions reported in previous literature has generally been in the range of 20 vol%–30.0 vol%, which is much lower than other oxide materials [19,21,22]. It has been found that coating is an effective method to alter the surface properties of the particles and is feasible to improve the stability and dispersity of the core material during colloidal processing [23,24]. Notably, several studies have shown that $ZrO_2$ possesses much higher inertness in aqueous suspension compared to $Y_2O_3$ [25,26]. In our previous research [10], particles consisting of a $Y_2O_3$ core coated with $ZrO_2$ were developed via the precipitation method by taking dual advantages of $ZrO_2$ as particle surface passivator and sintering aid, and high-transparency $ZrO_2$ doped $Y_2O_3$ ceramic was fabricated. Nevertheless, the coating process is complicated, and sintering still involves vacuum sintering.

In this work, a simple method to obtain $ZrO_2$-coated $Y_2O_3$ nanoparticles by thermal decomposition of zirconium nitrate was described to improve the dispersion of $Y_2O_3$ in aqueous suspensions. Highly transparent $Y_2O_3$ ceramics were prepared by a simple shaping approach. followed by air presintering plus HIPing without annealing. The effect of coating on the dispersity and stability of the $Y_2O_3$ suspensions during colloidal processing was investigated in comparison with uncoated powder.

## 2. Materials and Methods

### 2.1. Starting Materials

High-purity commercial $Y_2O_3$ nanopowder (99.99% pure; Huizhou Ruier Rare Chemical Hi-Tech Co., Ltd., Huizhou, China) was adopted as the starting material. Zirconium nitrate ($Zr(NO_3)_4 \cdot 5H_2O$, analytical reagent, Sinopharm chemical reagent, Shanghai, China) was adopted to prepare a 0.2 mol/L solution. Triammonium citrate (TAC) (Sinopharm Chemical Reagent Co., Ltd., Shanghai, China) was used as dispersant. The pH value of the suspensions was adjusted by HCl or KOH solution.

## 2.2. Coating Procedure and Suspension Preparation

$Y_2O_3$ powder was dispersed in ethyl alcohol with 2.0 wt.% dispersant added. After milling with $ZrO_2$ balls for 12 h, the suspension was then dried at 90 °C in an oven. The dried powder was sieved through a 200-mesh nylon screen. The powders were then calcinated at 1100 °C for 4 h, which was named starting powder (S.P.).

The S.P. was dispersed into ethyl alcohol, followed by the addition of a corresponding ratio of 5.0 at% $Zr^{4+}$ zirconium nitrate solution, and then the suspension was milled for 24 h. Subsequently, the suspension was dried at 80 °C for 12 h. The dried hygroscopic powder (D.P.) consisting of a complex hydrate of dried metal nitrate salt was consecutively calcined at 600, 700, and 800 °C for 4 h in a furnace to obtain $ZrO_2$-coated $Y_2O_3$ nanopowder. The mass ratio of TAC for different solid loadings of suspensions was 1.5 wt.%.

## 2.3. Consolidation and Sintering

The centrifugal slip-casting method was adopted using a laboratory centrifuge at 3000 rpm for 40 min. The green compacts were left at room temperature for 24 h, followed by drying at 80 °C for 12 h in an oven. The compacted samples were then presintered at 1500 °C for 16 h in air to reach the final stage of sintering. Finally, the presintered samples were post-HIPed at 1550 °C for 2 h under 200 MPa in argon atmosphere. Figure 1 shows a flowchart of the experimental procedure.

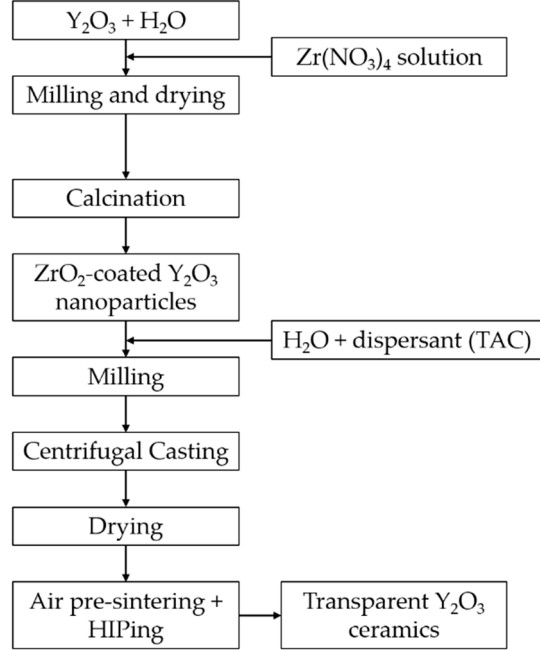

**Figure 1.** Flowchart for preparation of transparent $Y_2O_3$ ceramics.

## 2.4. Characterization

The crystal phase was identified by XRD (X'pert PROMPD, PANalytical, Almelo, the Netherlands) using nickel-filtered CuKα radiation. The morphologies of the powders were observed by transmission electron microscopy (TEM, Model JEM-2100F, JEOL, Tokyo, Japan). Zeta potential measurements were adopted to analyze the charged state of the particles' surface (DT-1202, Dispersion Technology Inc., New York, NY, USA). The rheological properties of the suspensions were tested by a cone-plate viscometer (Brookfield DV-II+Pro, Brookfield Engineering Laboratories, Middleboro, MA, USA). The relative densities of the green compact and sintered bodies were measured by the Archimedes method with regard to the theoretical density of $Y_2O_3$ (5.03 g/cm$^3$). The morphologies of the sintered bodies were observed by FE-SEM using a Hitachi scanning electron microscope (S-4800, Tokyo, Japan). The samples after polishing on both surfaces were used to measure the in-line

transmittance by using an ultraviolet/visible/near-infrared spectrophotometer (Model Lambda 750S, PerkinElmer, Waltham, MA, USA).

## 3. Results and Discussion

### 3.1. Coating $Y_2O_3$ Particle with Zirconium Nitrate via Thermal Decomposition Method

XRD patterns of the S.P., D.P., and powder after calcination at different temperatures are shown in Figure 2. All the peaks match the standard cubic $Y_2O_3$ phase (JCPDS card, No. 43-1036), and no $ZrO_2$ phase was detected. Compared with the S.P., the diffraction peak intensity of the D.P. was weakened, indicating the presence of zirconium nitrate crystal on the surface of the $Y_2O_3$ particle. The radius of $Zr^{4+}$ ions ($r = 0.72$ Å) is smaller than $Y^{3+}$ ions ($r = 0.89$ Å), which will shift the diffraction peaks to larger angles [27]. The diffraction peaks of the powders shift to larger angles after 600 °C calcination, indicating the dissolution of $Zr^{4+}$ within the $Y_2O_3$ lattice gradually with the decomposition of $Zr(NO_3)_4$. With a further increase in the calcination temperatures from 600 to 800 °C, the diffraction peaks of $Y_2O_3$ continuously shift towards larger angles (Figure 2b), which leads us to conclude that the increased dissolution of $ZrO_2$ within the $Y_2O_3$ lattice coincides with the increase in calcination temperature.

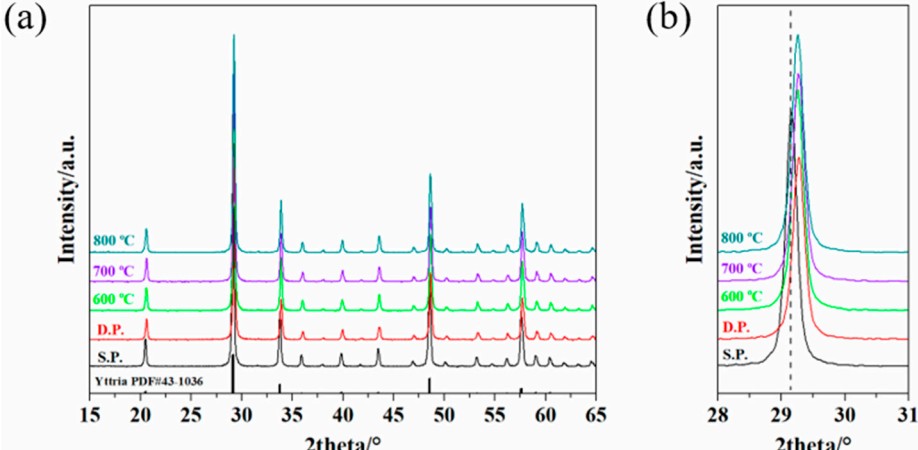

**Figure 2.** X-ray diffraction patterns of starting powder, dried powder, and $ZrO_2$-coated powders calcined at varied temperatures, (**a**) XRD patterns in the 2θ range of 15–65°, (**b**) Magnification of XRD patterns in the 2θ range of 28°–30°.

In order to deeply investigate the evolution during the coating procedure, the morphological characteristics of the powders were analyzed by transmission electron microscopy, as shown in Figure 3. The particles for the S.P. possess high crystallinity with flat and clear edges. The average size of the particles is ~100.0 nm (Figure 3a). In contrast, the D.P. exhibits rough surfaces with loosely packed tiny particles of zirconium nitrate crystal, as shown in Figure 3b. On account of that, the S.P. was already calcined at 1100 °C before the coating process, and there was no significant change in the morphology of the core particle after different temperature calcination. However, it is worth noting that with the increasing calcination temperature, the tiny particles consisting of the coating layer gradually decreased and disappeared (Figure 3c,d,f). As we can see from Figure 3d,e, after calcination at 700 °C, a coating layer with a thickness of 5–10 nm could be observed. On the one hand, the d-spacing of the coating layer of 2.94 Å could be indexed to the $ZrO_2$ (tetragonal) structure of the (011) crystal plane. On the other hand, the d-spacing of 5.29 Å is the (200) plane for the $Y_2O_3$ (cubic) structure. With the further increase in calcination temperature to 800 °C (Figure 3f), the coating layer nearly disappeared, indicating that $ZrO_2$ gradually dissolved into the $Y_2O_3$ lattice with the increase in calcination temperature.

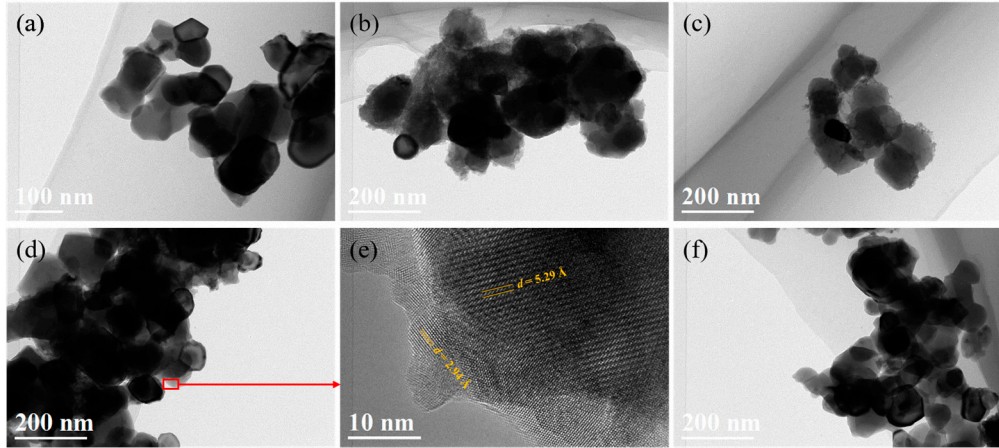

**Figure 3.** TEM images and corresponding HRTEM images of starting powders, (**a**), dried powders (**b**), and powder with ZrO$_2$ coating layer calcined at different temperatures of 600 °C (**c**), 700 °C (**d**,**e**), and 800 °C (**f**).

During calcination, Zr(NO$_3$)$_4$ decomposed and adhered onto the surfaces of Y$_2$O$_3$ particles. ZrO$_2$ dissolved into the Y$_2$O$_3$ lattice with the increase in calcination temperature. The diffusion depth of Zr$^{4+}$ into Y$_2$O$_3$ lattice for the sample calcinated at 700 °C for 4 h was evaluated using the following equations [28]:

$$D_{\text{lattice}} = \exp(-423 \, (\text{kJ/mol}) / RT) \, \text{m}^2\text{s}^{-1} \tag{1}$$

where $D_{\text{lattice}}$ is the diffusion coefficient, and $T$ is temperature. According to Boniecki's research, the diffusion coefficient of Zr$^{4+}$ in the Y$_2$O$_3$ lattice at 1600 °C was $7.04 \times 10^{-22}$ m$^2$s$^{-1}$; hence, we can calculate the diffusion coefficient of Zr$^{4+}$ in the Y$_2$O$_3$ lattice at 700 °C as follows [29]:

$$D_{\text{lattice}}^{700} = \frac{\exp(-423 \, (\text{kJ/mol}) / \text{Rg}927) \, \text{m}^2\text{s}^{-1}}{\exp(-423 \, (\text{kJ/mol}) / \text{Rg}1827) \, \text{m}^2\text{s}^{-1}} \times 7.04 \times 10^{-22} \tag{2}$$

where Rg is the ideal gas content, and 927 and 1827 are the absolute temperatures of 700 and 1600 °C. As the holding time of calcination was 4 h, the diffusion area was achieved as follows:

$$\text{S} = D_{\text{lattice}}^{700} \times 4 \times 3600 \approx 9.30 \, \text{nm}^2 \tag{3}$$

where $S$ is the diffusion area. The diffusion depth could then be evaluated as 3.44 nm, which was far below the radius of the raw particles (100 nm). On the basis of our findings, one can conclude that powder after 700 °C calcination consisted of surface-coated ZrO$_2$ and Zr$^{4+}$ solid solution outside of the Y$_2$O$_3$ lattice, which was beneficial to the inhibition of hydrolysis issue.

### 3.2. Dispersion Properties of ZrO$_2$-Coated Y$_2$O$_3$ Powders

Figure 4 shows the rheological behaviors of suspensions for the S.P. and coated powders with a solid loading of 10.0 vol% at pH = 10.1 with TAC added. The viscosity of the suspension with S.P. (S-suspension) decreases as the shear rate increases, showing a shear-thinning behavior. For the suspension with coated powder calcinated at different temperatures, there was a significant decrease in the viscosity, which exhibits a relatively constant viscosity corresponding to Newtonian. The lowest viscosity occurred when the calcination temperature was 700 °C. According to the above analysis, suspensions formulated with powders calcined at 700 °C (C-suspension) were adopted to be a better choice for the following steps to fabricate transparent ceramics.

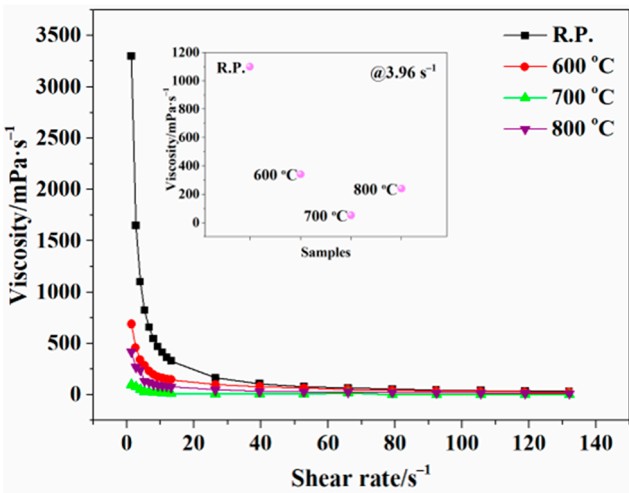

**Figure 4.** Viscosity for 10.0 vol% suspensions of powders calcinated at different temperatures, inset picture: viscosity for suspensions @3.96 s$^{-1}$.

The zeta potential of the $ZrO_2$-coated $Y_2O_3$ powders calcined at 700 °C for 4 h calcination, the S.P., and the synthesized $ZrO_2$ powder treated under identical conditions are shown in Figure 5. The isoelectric point (IEP) of the S.P. and synthesized $ZrO_2$ powder are 10.1 and 8.0, respectively. The IEP for $ZrO_2$-coated $Y_2O_3$ powders showed between the S.P. and synthesized $ZrO_2$ powder. The coating of $ZrO_2$ reduced the IEP of $Y_2O_3$ particles from 10.1 to 9.3. The negatively charged groups of TAC were absorbed on the surfaces of $ZrO_2$-coated yttria nanoparticles. As observed from this figure, the addition of TAC shifted the IEP value from 10.1 to 5.6. Moreover, the absolute zeta potential of particles in suspensions increases with the increase in pH, and it reaches 49 mV at the pH of 11. The good dispersion of coated $Y_2O_3$ powder in aqueous suspension is favorable for the preparation of well-dispersed and high solids loading solution in an easy method.

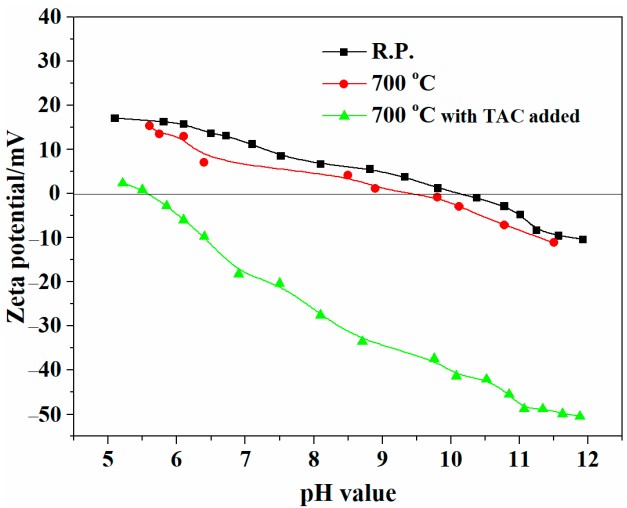

**Figure 5.** Zeta potential versus pH for 10.0 wt.% suspensions with different powders and TAC added.

### 3.3. Consolidation and Sintering of $Y_2O_3$ Transparent Ceramics

Figure 6 shows the SEM images of the green bodies consolidated from two suspensions with a solid loading of 35.0 vol%. Larger micron-scale pores can be observed for the sample consolidated using S-suspensions (Figure 6a), which are difficult to eliminate during the sintering process. In sharp contrast, the green bodies of the C-suspension yielded a more homogeneous and compact microstructure (Figure 6b). Only nanosized pores can be seen, and the particles are in close contact with each other. The relative densities of the green bodies are 50.2% and 29.2% for the C-suspension and S-suspension, respectively.

As we all know, green compacts with smaller pore size distribution and a homogeneous microstructure show high sinterability and were easy to densify at lower temperatures.

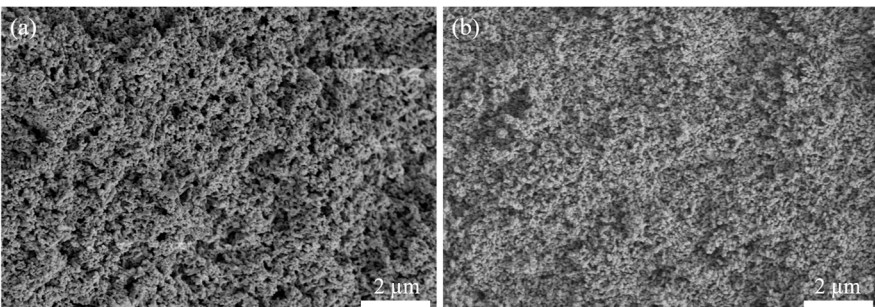

**Figure 6.** SEM micrographs of the fracture surfaces of the green compacts were obtained from (**a**) S-suspension and (**b**) C-suspension.

Figure 7 shows SEM images of the $Y_2O_3$ ceramics sintered at 1500 °C for 16 h in air. For the sample prepared from the S-suspension, a porous structure with the presence of open pore channels is shown in Figure 7a. The relative density of the sample is 86.4%, which is the typical feature of intermediate-stage sintering [30]. In comparison, for the sample prepared from the C-suspension (Figure 7b), the number of pores is much less than that in the S-suspension one. Moreover, it can also be noticed that the sample has a much smaller average pore size. The relative density of the sample reaches 96.4%, and there are no open pores found in the sample, suggesting that it is already at the final stage of sintering [27]. From the results we obtained, one can conclude that the higher relative density and more homogeneous microstructure improved the sinterability.

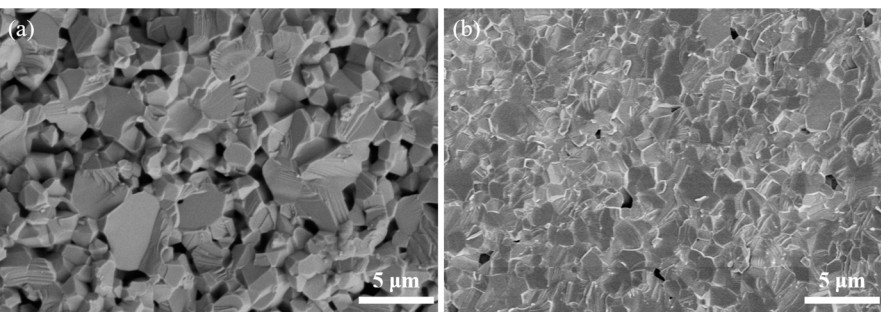

**Figure 7.** SEM micrographs of sintered bodies for 16 h at 1500 °C obtained from (**a**) S-suspension and (**b**) C-suspension.

The in-line transmittance curves of the samples are shown in Figure 8. The transmittance of the sample prepared by the S.P. is lower than its theoretical value. The relative density of the sample is 95.5%. In contrast, the sample prepared with coated powder after 700 °C calcination has a remarkable improvement in transmittance observed. The relative density of the sample is 99.9%. The in-line transparency (ILT) at 1100 nm is 81.4%. At the wavelength of 2000 nm, the ILT approaches 82.6%, which is very close to the theoretical value of $Y_2O_3$ crystal. The inset photograph shows the optical images of the as-obtained $Y_2O_3$ ceramic prepared with coated powder after polishing, and the characters below the ceramic sample can be seen very clearly.

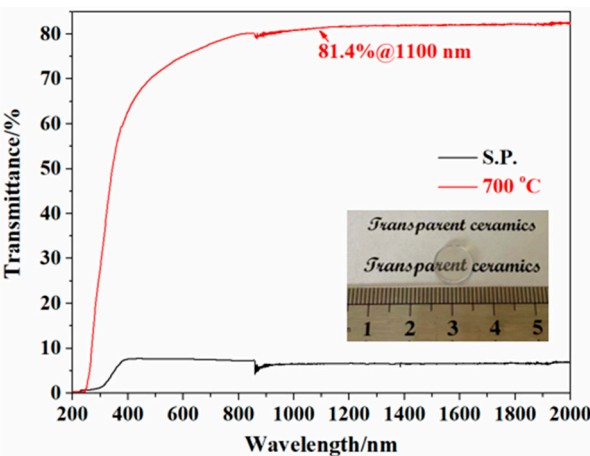

**Figure 8.** In-line transmittance of HIPed $Y_2O_3$ ceramics at 1550 °C (thickness = 1.0 mm), inset photograph: ceramic prepared with coated powder after polishing.

It is worth mentioning that the absorption edge shifts to the slong wavelength of the sample prepared by the S.P. at the wavelength of 250–350 nm range could be observed. The addition of $Zr^{4+}$ to $Y_2O_3$ will produce oxygen interstitial of $O_i''$ during the sintering process and inhibit the grain boundary mobility to promote the densification of $Y_2O_3$ ceramics. Herein, the shift of the undoped sample could be due to the change of band gap energy and the oxygen vacancies [31] formed during the sintering process. Moreover, the very tiny little residual pores in the sintered body could also be an important issue for the decrease in the transmittance at short wavelengths.

Figure 9 shows the microstructure of the fracture surfaces for the HIPed samples. Compared with the sample prepared by the C-suspension, the sample prepared by the S-suspension (shown in Figure 9a) exhibited an exaggerated grain size and lots of intergranular pores and intragranular pores act as light scattering centers in transparent ceramics, resulting in a decrease in transmittance. The average grain size of the sample obtained from the S.P. is 3.3 μm. However, fully densified and no pores are observed in the sample prepared by $ZrO_2$-coated powder (Figure 9b,c), which helps to explain their higher optical quality. The typical grain size distribution is shown in Figure 9d. The mean grain size is 1.7 μm, indicating that $ZrO_2$ effectively promoted the densification and inhibited the grain growth of the sintered body.

Compared with Wang and Liu's results [7,8], we achieved highly transparent $Y_2O_3$ ceramic with fine grain size, and the ILT was ~80.0% in the visible wavelength range by air presintering plus HIPing at a lower temperature. Herein, we proposed a new method to synthesize $ZrO_2$-coated $Y_2O_3$ powder for colloidal processing. By adopting $ZrO_2$-coated $Y_2O_3$ powder, a well-dispersed suspension with high solid loading (35.0 vol%) was achieved. After consolidation, the green compacts were sintered in an energy-saving and effective method, which is low-temperature air presintering plus HIPing without annealing. Moreover, the sintered body was fully densified, and the average grain size is less than 2.0 μm, which may make it a promising fabrication method in the preparation of sesquioxide transparent ceramics. We point out that this is the most effective and simple method to fabricate highly transparent ceramics with fine grain size.

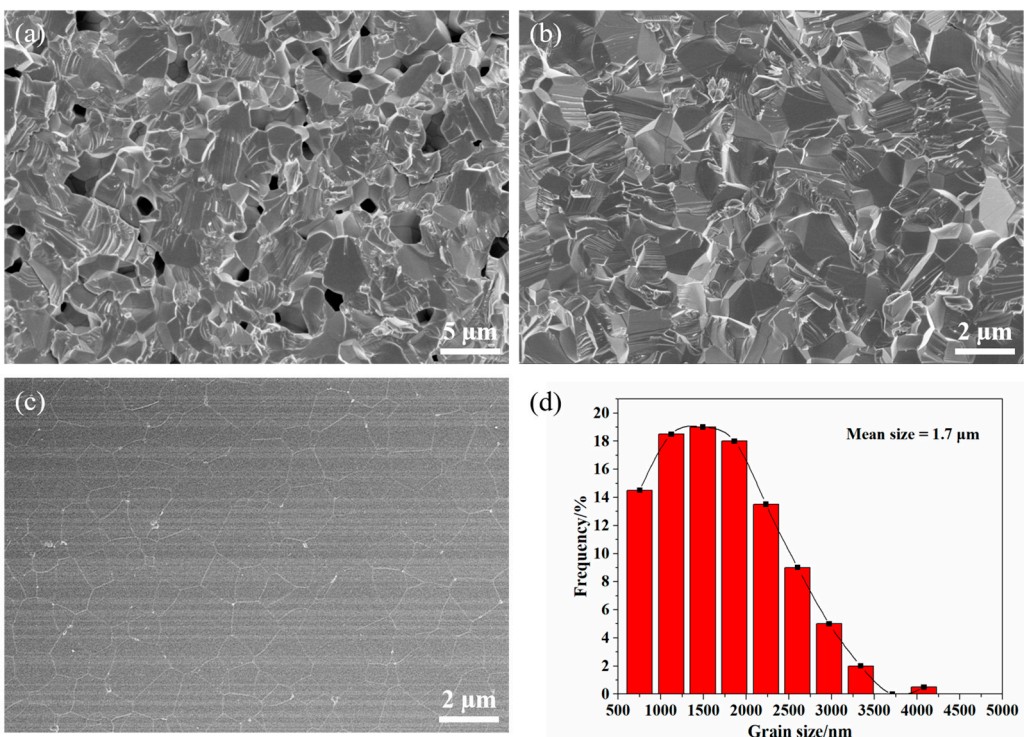

**Figure 9.** SEM micrographs of the fracture surfaces of HIPed $Y_2O_3$ ceramics obtained from (**a**) S.P. and (**b**) $ZrO_2$ coated powder after 700 °C calcination, (**c**) thermally etched surfaces of sample obtained from $ZrO_2$ coated powder after 700 °C calcination and (**d**) grain size distribution of the sample.

### 4. Conclusions

In summary, the coating of $Y_2O_3$ particles with a thin $ZrO_2$ layer via an easy method was reported, and it was found to be an effective method to inhibit hydrolysis issues during suspension preparation. Consequently, a homogeneous green body with a relative density of 50.2% was obtained, which is beneficial for fabrication of highly transparent $Y_2O_3$ ceramics with fine grain size via low-temperature air presintering plus HIPing treatment. The $Y_2O_3$ ceramic presintered at 1500 °C × 16 h and post-HIPed at 1550 °C × 2 h showed a grain size of 1.7 μm and high in-line transmittance (81.4% at 1100 nm, thickness: ~1.0 mm), very close to the theoretical value of $Y_2O_3$. This is the most effective and simple method to fabricate highly transparent ceramics with fine grain size.

**Author Contributions:** Writing—original draft preparation, data curation, formal analysis, and investigation, Z.F.; data curation, H.L. writing—review and editing, X.L. supervision, N.W., J.W., J.Z., Z.H. and X.S. All authors have read and agreed to the published version of the manuscript.

**Funding:** This research was funded by the postdoctoral research start-up funding of Shenyang University (No. 1220502052022010215).

**Institutional Review Board Statement:** Not applicable.

**Informed Consent Statement:** Not applicable.

**Data Availability Statement:** Data sharing is not applicable to this article.

**Conflicts of Interest:** The authors declare that they have no known competing financial interests or personal relationships that could have appeared to influence the work reported in this paper.

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
