# Peer review of "Fabrication of Highly Transparent Y2O3 Ceramics via Colloidal Processing Using ZrO2-Coated Y2O3 Nanoparticles"

_coatings, doi:10.3390/coatings12081077_

Round 1

Reviewer 1 Report

Please see the enclosed file.

Reviewer 2 Report

This is a fairly solid work, which should undoubtedly be recommended for publication, but after clarifying some incomprehensible points.

1.     Line 28-32. Somewhere in here information about the band gap energy for Y2O3 should be given. See for details: Table 1 (for Y2O3 Eg=6.3 eV)  in paper:

Mechanisms of intrinsic and impurity luminescence excitation by synchrotron radiation in wide-gap oxides,  Kirm, M., Feldbach, E., Kink, R., Maaroos, A., Martinson, I.

Journal of Electron Spectroscopy and Related Phenomena, 1996, 79, pp. 91–94.

2.     Line 51. Important information is not noted here, namely that the structural interaction of Y2O3 and ZrO2 leads to the stabilization of YSZ with several unique properties. See:  Savoini, B.; Ballesteros, C.; Santiuste, J.M.; González, R.; Popov, A.I.; Chen, Y. Copper and iron precipitates in thermochemically reduced yttria-stabilized zirconia crystals. Philos. Mag. Lett. 200181, 555–561

3.     Line 126-129. Small concentrations of ZrO2 may not be visible by diffraction but can be easily seen by fluorescent spectroscopy.

4.     Fig.7. Transmission data needs more detailed discussion. It is easy to see that the absorption edge is not at 6.3 eV, as in single crystals. It is however well known that point defects are formed in almost all types of synthesis. Don't you think that the observed shift of the absorption edge at 250-350 nm can be due, as a many oxides, including similar Al2O3, can be due to oxygen vacancies?  Here it is important to emphasize once again that the formation of  such defects (F-type centers/ oxygen vacancies with trapped electrons) on the long-wavelength tail of the absorption edge is a well-known fact not only in single crystals, but also in nanomaterials.  See, for example, MgAl2O4 (crystals and ceramics):  see Fig.1.  in Lushchik, A., Feldbach, E., Kotomin, E.A. et al. Distinctive features of diffusion-controlled radiation defect recombination in stoichiometric magnesium aluminate spinel single crystals and transparent polycrystalline ceramics. Sci Rep 10, 7810 (2020). https://doi.org/10.1038/s41598-020-64778-8

Reviewer 3 Report

This is a good article that should definitely be recommended for publication, but some points within manuscript need clarification.

1.     Since the article deals also with optical properties, it would be appropriate to give the band gap energy data in the introduction.

2.     How does the interaction of Y2O3 and ZrO2 occur, what new phases could be formed?

3.     Actually, luminescence studies of such structures were very useful.

4.     The transmittance of HIP-coated Y2O3 ceramics, especially the transmittance in the short-wavelength part of the spectrum, requires an explanation in terms of band gap change or increased defectiveness, as shown in Fig.7.

Reviewer 4 Report

Most of the references are very old. This is a very old system and thousands of papers have been published on the combined material. There is not much novelty in this work. Please look into the literature carefully. 

The author should be asked to compare the results with the new results in the field. This comprises all the data like transparency and the electrical properties of the compound with the literature.

Round 2

Reviewer 2 Report

Revised version is suitable for publication and van be published as it is.

Author Response

Thank you for your kindness.

Reviewer 3 Report

The authors responded to all reviewer comments, so paper can be recommended for publication.

Author Response

Thank you for your kindness.

Reviewer 4 Report

My previous comments were 

Point 1: Most of the references are very old. This is a very old system and thousands of papers have been published on the combined material. There is not much novelty in this work. Please look into the literature carefully.

The author says they have replaced the old references with new ones. It is not a matter of replacing. Reference is given from where you have taken the matter; it is not whimsical. I meant that you have to refer to the recent literature and update the values and ad the new development in the field.

Point 2: The author should be asked to compare the results with the new results in the field. This comprises all the data like transparency and the electrical properties of the compound with the literature.

They also missed this point; I am asking you to quote the values reported by others and then compare them with your results and discuss scientifically the reason for the difference between the value. Without citing others' values, just saying that our is the highest is not scientific. 
